# Complement System Activation Is a Plasma Biomarker Signature during Malaria in Pregnancy

**DOI:** 10.3390/genes14081624

**Published:** 2023-08-14

**Authors:** Veronica Feijoli Santiago, Jamille Gregorio Dombrowski, Rebeca Kawahara, Livia Rosa-Fernandes, Simon Ngao Mule, Oscar Murillo, Thais Viggiani Santana, Joao Victor Paccini Coutinho, Janaina Macedo-da-Silva, Lucas Cardoso Lazari, Erika Paula Machado Peixoto, Marcel Ivan Ramirez, Martin R. Larsen, Cláudio Romero Farias Marinho, Giuseppe Palmisano

**Affiliations:** 1Department of Parasitology, Institute of Biomedical Sciences, University of São Paulo, São Paulo 05508-000, Brazil; 2Analytical Glycoimmunology Group, Department of Molecular Sciences, Macquarie University, Macquarie Park, NSW 2109, Australia; 3Department of Pulmonary Immunology, Center for Biomedical Research, University of Texas Health Center Science at Tyler, Tyler, TX 75708, USA; 4Cell Biology Laboratory, Carlos Chagas Institute, Fiocruz, Curitiba 81350-010, Brazil; 5Department of Biochemistry and Molecular Biology, University of Southern Denmark, 5230 Odense, Denmark

**Keywords:** malaria, malaria in pregnancy, *Plasmodium berghei*, *Plasmodium falciparum*, complement system, biomarkers, plasma proteomics

## Abstract

Malaria in pregnancy (MiP) is a public health problem in malaria-endemic areas, contributing to detrimental outcomes for both mother and fetus. Primigravida and second-time mothers are most affected by severe anemia complications and babies with low birth weight compared to multigravida women. Infected erythrocytes (IE) reach the placenta, activating the immune response by placental monocyte infiltration and inflammation. However, specific markers of MiP result in poor outcomes, such as low birth weight, and intrauterine growth restriction for babies and maternal anemia in women infected with *Plasmodium falciparum* are limited. In this study, we identified the plasma proteome signature of a mouse model infected with *Plasmodium berghei* ANKA and pregnant women infected with *Plasmodium falciparum* infection using quantitative mass spectrometry-based proteomics. A total of 279 and 249 proteins were quantified in murine and human plasma samples, of which 28% and 30% were regulated proteins, respectively. Most of the regulated proteins in both organisms are involved in complement system activation during malaria in pregnancy. CBA anaphylatoxin assay confirmed the complement system activation by the increase in C3a and C4a anaphylatoxins in the infected plasma compared to non-infected plasma. Moreover, correlation analysis showed the association between complement system activation and reduced head circumference in newborns from *Pf*-infected mothers. The data obtained in this study highlight the correlation between the complement system and immune and newborn outcomes resulting from malaria in pregnancy.

## 1. Introduction

Malaria during pregnancy can have serious consequences, especially in primigravida and second-time mothers. According to the World Health Organization, in 2021, approximately 13.3 million pregnant women in the African region were exposed to the risks of Malaria in pregnancy (MiP) [1]. Pregnant women are included in the group most susceptible to malaria infection since infected erythrocytes (IE) accumulate in the maternal intervillous space of the placenta. *Plasmodium falciparum* (*P. falciparum*) IE binds to Chondroitin Sulfate A (CSA) through the *P. falciparum* erythrocyte membrane protein 1 (PfEMP1) variant known as VAR2CSA [2,3]. Upon binding, monocyte activation with inflammatory cytokines production lead to maternal anemia and intrauterine growth restriction [4,5]. Syncytiotrophoblasts are responsible for the secretion of beta-chemokine (monocyte chemotactic protein-1, MCP-1) and macrophage inflammatory protein-1α and β. Monocytes infiltrate into the placental tissue [6] and mediate inflammation through the production of TNFα, IFNγ, IL-1β, and proteins from the complement system [7]. 

The complement system is part of the innate immune response targeting foreign cells and communicating with the adaptive immune response [8]. The complement system is deeply regulated by fluid-phase complement proteins such as Factor H, C4b-binding protein, C1-inhibitor, and Factor H-like protein 1. Moreover, the complement system is also controlled by cell membrane proteins, for example, decay-accelerating factor (DAF)/CD55, protectin/CD59, complement receptor 1 (CR1), and membrane cofactor protein (MCP/CD46) [9]. In uncomplicated pregnancies, complement system activation is involved in placental development in several stages such as trophoblast invasion of decidua [10,11]. Additionally, trophoblast cells express complement regulatory proteins (DAF/CD55), MCP/CD46, and CD59 to protect against fetal injury, preventing dysregulated complement activation [12]. Regulation of the complement system is tightly controlled during pregnancy; however, excessive complement activation is harmful to a developing fetus [13,14]. The excessive activation of the complement system by the malaria infection is induced through all three complement pathways [15]. Components of the classical pathway, such as C1r, C1s, and C1inh, are elevated during severe malaria [16]. Moreover, the C1q component was identified as upregulated in the placentas of primigravida women infected by *Plasmodium falciparum* [13]. Regarding the alternative pathway, hematin (erythrocyte lysis product) and platelet-derived Factor D are able to activate the alternative pathway [17]. The role of the complement system as a mediator of adverse outcomes during malaria in pregnancy contributes to tissue injury and immunopathology. The excessive expression of pro-inflammatory C5a and/or Bb fragments has been associated with poor MiP outcomes [18,19,20]. Indeed, human monocytes exposed to *Plasmodium falciparum* glycosylphosphatidylinositol (*Pf*GPI) induce the expression of C5aR, and the increased levels of C5a-C5aR and TLR2-*Pf*GPI signaling enhance monocyte recruitment and inflammation in the placenta [14]. Infiltrated monocytes and macrophages secrete cytokines (such as TNFα) and are associated with MiP poor outcomes [21]. Moreover, malaria in pregnancy is related to an imbalance of angiogenic factors (VEGF, sFlt1, soluble endoglin, and angiopoietins), which results in early gestational age and low birth weight [15,22,23]. Increased levels of the complement anaphylatoxin C5a correlate with angiogenic factors alterations. Additionally, C5a induces monocyte secretion of anti-angiogenic factor sFlt1 [21]. These data suggest that complement system dysregulation is associated with altered angiogenesis and placental inflammation, leading to adverse birth outcomes during MiP.

The pathological alterations in the placenta compromise maternal–fetal nutrient and oxygen exchange, resulting in intrauterine growth retardation and low birth weight [23]. Although several tools have been introduced to diagnose malaria during pregnancy, there are few or no specific markers to evaluate placental dysfunctions during MiP and their influence on the fetus [23,24,25,26]. Thus, studying the pathophysiology of MiP in an in vivo animal model and patients constitutes a powerful approach in identifying molecular traits to elucidate new mechanisms of disease pathogenesis and potential biomarkers. We have implemented a translational approach to identify plasma proteins associated with malaria during pregnancy. The plasma proteome of *P. berghei*-infected pregnant mice was initially analyzed using quantitative mass spectrometry-based proteomics. This murine model was chosen since it could recapitulate several pregnancy outcomes observed in human infections, such as prominent inflammatory response, reduced fetal viability, and intrauterine growth retardation [27]. Subsequently, the same analytical platform was used to analyze the plasma proteome of *P. falciparum*-infected pregnant women. Differential expression of complement system components was detected during MiP and correlated with immunological and clinical parameters.

## 2. Materials and Methods

### 2.1. Mice and Parasites Strains

BALB/c female mice with 8 to 10 weeks (n = 4 per group) were bred and maintained in conventional housing with a constant light–dark cycle (12 h:12 h) at the Animal Facility of the Department of Parasitology of the Institute of Biomedical Sciences at the University of São Paulo (ICB/USP). All experiments were performed under the ethical guidelines for experiments with mice, and the Animal Health Committee of the Institute of Biomedical Sciences of the University of São Paulo approved the protocols (CEUA No n° 47, fls. 07, book 03). The guidelines for animal use and care were based on the standards established by the National Council for Control of Animal Experimentation (CONCEA).

The murine model of MiP was developed based on the study of Marinho et al. [28]. The *Plasmodium berghei* (*P. berghei*) strain used in the experiment was *P. berghei* ANKA^GFP/HSP70^ (constitutively expressing GFP in association with the Hsp70 protein), provided by Dr. Rogério Amino (Malaria infection & Immunity Laboratory, Pasteur Institute, Paris, France). Before infection, a donor BALB/c mouse was infected via intraperitoneal (i.p.) injection with 10^6^ IE obtained from frozen stock. Parasites were maintained by serial passage of blood until the parasitemia reached 5–10%. A donor mouse was sacrificed, and blood was collected via cardiac puncture. A schematic representation of *P. berghei* ANKA^GFP/HSP70^ infection of mice is reported in Figure 1. Each BALB/c female mouse used in the experiment (n = 4) was infected i.p. with 10^4^ IE and treated with 0.7 mg chloroquine 7 days post-infection for 3 days. At forty days post-infection, a group of treated females was mated with BALB/c males. The same procedure was performed for the uninfected group (n = 4). Vaginal plug detection and weight measurement were used to monitor the pregnancy time, as previously described [27]. The pregnancy was confirmed between the 10th and 13th gestational days by the weight gain of 3–4 grams. On the 19th gestational day, both infected pregnant females and control pregnant females were euthanized, and blood was collected into a vacuum tube containing citrate. Plasma was isolated using centrifugation at 1500× *g* for 15 min at 4 °C.

### 2.2. Bodyweight and Parasitemia Measurement during Infection

Body weight was measured daily starting on day 0 (the day of infection). In addition, parasitemia was measured daily in the tail blood with flow cytometry analysis using GFP constitutively expressed by the parasites (Appendix A). To perform the parasitemia measurement, 2 µL of blood was collected and added to 400 µL staining buffer (Phosphate Buffered Saline—PBS 1X, 2% of Fetal Bovine Serum, 0.05% sodium azide). Samples were analyzed in a FACSCalibur cytometer (BD Biosciences, Franklin Lakes, NJ, USA) and then analyzed using FlowJo software version X 10.0.7r2.

### 2.3. Mouse Cytokine Measurements Using Cytometric Bead Array

Cytokines IL-10, TNFα, IL-6, and IL-12p70 levels in mouse plasma samples were determined using Cytometric Bead Array Mouse Inflammation Kit Assay (BD Biosciences, USA) according to the manufacturer’s recommendations. Briefly, capture beads were incubated with the samples or the standard (standard curve values ranged from 20 pg/mL to 5000 pg/mL) and the detection reagent. Samples and standards were incubated for 2 h at room temperature and protected from light. Beads were washed and centrifuged at 270× *g* for 3 min to remove unbounded detection antibodies in the supernatant. Beads were resuspended in a wash buffer, quantified using flow cytometry (FACSCalibur, BD Biosciences, USA), and analyzed with FCAP Array software version 3.0 (BD Biosciences, USA). Cytokine concentration (pg/mL) was calculated with a standard regression curve. Statistical test Mann–Whitney test (*p* < 0.05) was performed to compare infected and control groups using GraphPad Prism 6.0. 

### 2.4. Birth Parameters, Vascular Space, and Spleen Weight Measurements

Murine spleen, fetal and placental weight, and vascular space measurements were obtained on the 19th day of pregnancy (Appendix A). Placental histological and morphometric analyses were performed as described by Neres et al. [28]. Briefly, for vascular space quantification, placental tissue sections were stained with hematoxylin–eosin (H&E). Each section had three regions randomly selected, and images were taken using the Zeiss camera (Axio Cam HRc, ZEISS group, Jena, Germany) connected to a Zeiss light microscope (Axio imager. M2, ZEISS group, Jena, Germany). After image acquisition, ImageJ software was used to perform morphometric analysis. A limit of colors to images that covered the areas corresponding to vascular space was stipulated for each picture. The percentage of area covered was measured for the ratio between the number of pixels of the defined area and the total pixels of the image. The vascular area of each placenta was estimated by analysis of two sections in a non-successive way.

### 2.5. Study Design of Human Malaria in a Pregnancy Cohort

Pregnant women (non-infected and infected during pregnancy) between January 2013 and April 2015, a total of 600 from an endemic Brazilian region in the state of Acre (Alto Juruá Valley), were enrolled through volunteer sampling and followed-up until delivery. Socioeconomic and clinical data were collected, and peripheral blood was used to diagnose and/or confirm malaria infection using microscopy and real-time PCR. At delivery, clinical data were collected from the mother and newborn, as well as placental tissue and blood samples. According to the Brazilian Ministry of Health guidelines, all pregnant women positive for malaria were treated with antimalarial drugs under medical prescription. Exclusion criteria involved: (1) history during pregnancy of smoking and/or alcohol consumption, (2) other infections (i.e., Toxoplasmosis, HIV, Hepatitis B and C viruses, and Syphilis), (3) other comorbidities (i.e., hypertension, preeclampsia/eclampsia, and diabetes mellitus), and (4) preterm delivery, stillbirth, and newborn with congenital malformation. After applying the exclusion criteria, samples were selected between uninfected pregnant women and *Plasmodium falciparum-infected* women during pregnancy. A schematic representation of the study design and women selection is reported in Appendix A. For the proteomic analysis, a total of 18 pregnant women, 9 control and 9 infected by *P. falciparum*, were selected. Clinical and epidemiological data (gestational age, gravidity, age, etc.) of the infected and uninfected pregnant women (control group) were matched. Cytokine levels (IL-10, IL-6, IL-12, TNF-α) from maternal peripheral plasma were measured. Additionally, the clinical parameters of newborns were also measured by trained obstetrician nurses. Head and chest circumferences, as well as newborn length, were expressed in centimeters using a non-stretching flexible tape. Moreover, Rohrer ponderal index (RI) was calculated as described by Dombrowski et al. [29]. Briefly, this parameter was calculated based on the newborn’s weight (g) divided by the cube of the length (cm) (Appendix A). Ethical clearance was provided by the Ethics Committee in Research involving Human Beings of the Biomedical Sciences Institute of São Paulo University under the Certificate of Presentation to the Ethic Appreciation (CAAE) number 03930812.8.0000.5467 and 32707720.0.0000.5467, according to Resolution no. 466/12 of the Brazilian National Health Committee.

### 2.6. Blood Sample Collection and Plasma Isolation

Peripheral blood was collected at delivery into vacuum tubes containing heparin. Plasma was isolated using centrifugation at 1200× *g* for 10 min at 4 °C and kept at −80 °C until the proteomics analysis.

### 2.7. Plasma Protein Extraction and Digestion for Mass Spectrometry

A total of 10 µL of plasma proteins were denatured in 8M urea, reduced with DTT (10 mM final concentration) for 30 min at room temperature, and subsequently alkylated with IAA (40 mM final concentration) for 30 min in the dark. Protein quantification was assessed using Qubit 3.0 Fluorometer (Invitrogen Thermo Fisher, Waltham, MA, USA). An amount of 100 µg of protein sample was analyzed for each sample. Then, the urea concentration was diluted to less than 1M to perform trypsin digestion at a 1:50 enzyme to protein ratio, overnight at 30 °C. After tryptic digestion, samples were acidified with TFA 1% final concentration, and digested peptides were desalted using C18 microcolumns. Peptides were resuspended in 0.1% Formic Acid (FA) and analyzed using LC–MS/MS.

### 2.8. Mass Spectrometry Analysis

Tryptic peptides derived from mouse plasma samples were analyzed into LTQ Orbitrap Velos ETD (Thermo Scientific, Waltham, MA, USA) coupled with Easy NanoLC II (Thermo Scientific, Waltham, MA, USA). The peptides were resuspended in 0.1% FA, injected, and loaded on ReproSil-Pur C18 AQ (Dr. Maisch, Ammerbuch-Entringen, Germany) in-house packed trap column (2 cm × 100 μm, inner diameter 5 μm). Peptides were separated at a flow of 300 nL/min on an analytical ReproSil-Pur C18 AQ (Dr. Maisch, Ammerbuch-Entringen, Germany) packed in a house column (17 cm × 75 μm × 3 μm) using reversed-phase chromatography, which was operated on an EASY NanoLC II system (Thermo Scientific). The mobile phase was water/0.1% FA (A) and 95% ACN/0.1% FA (B). The gradient was 0–45% of solvent B for 100 min. The EASY NanoLC II was coupled into an LTQ Orbitrap Velos mass spectrometer (Thermo Scientific) operating in positive ion mode. The mass spectrometer acquired a full MS scan at 60,000 full-width half-maximum (FWHM) resolution with a 350–1500 Da mass range. The top 20 most intense peptide ions were selected from MS for Collision Ion Dissociation (CID) fragmentation (normalized collision energy: 35.0 V).

Tryptic peptides derived from human plasma samples were analyzed in LTQ Orbitrap Velos ETD (Thermo Scientific, Waltham, MA, USA) coupled with Easy NanoLC II (Thermo Scientific, Waltham, MA, USA). Peptides were resuspended in 0.1% FA, injected, and loaded on ReproSil-Pur C18 AQ (Dr. Maisch, Ammerbuch-Entringen, Germany) in-house packed trap column (2 cm × 100 μm inner diameter 5 μm). Peptides were separated at a flow of 300 nL/min on an analytical ReproSil-Pur C18 AQ (Dr. Maisch, Ammerbuch-Entringen, Germany) packed in a house column (17 cm × 75 μm × 3 μm) using reversed-phase chromatography, which was operated on an EASY NanoLC II system (Thermo Scientific). The mobile phase was water/0.1% FA (A) and 95% ACN/0.1% FA (B). The gradient was 2–30% of solvent B for 80 min, 30–95% of solvent B for 5 min, and 95% B for 20 min. EASY NanoLC II was coupled into an LTQ Orbitrap Velos mass spectrometer (Thermo Scientific) operating in positive ion mode. The mass spectrometer acquired a full MS scan at 60,000 full-width half-maximum (FWHM) resolution with a 350–1500 Da mass range. The top 21 most intense peptide ions were selected from MS for Collision Ion Dissociation (CID) fragmentation (normalized collision energy: 35.0 V).

### 2.9. Database Search and Statistical Analysis

Raw files from mouse and human samples were searched using Proteome Discoverer software version 2.3 (Thermo Fisher) computational platform using the Sequest search engine. Raw data obtained from the mice samples were searched against *Mus musculus*-reviewed databases (June 2019 release; 17,034 entries). Raw data from human samples were searched against the *Homo sapiens* reviewed databases (July 2019 release; 20,352 entries). The parameters used were full trypsin specificity; two missed cleavages allowed, common contaminants, and reverse sequences including carbamidomethylation of cysteine as a fixed modification; and methionine oxidation and protein N-terminal acetylation as variable modifications. Identifications of peptide spectrum matches (PSMs), peptides, and proteins were accepted with less than a 1% false discovery rate (FDR). To identify proteins differentially regulated between the conditions, peptide group results were filtered based on the following parameters: (1) PSMs and peptide FDR below 1%, (2) no missed cleavages, (3) no methionine oxidation, and (4) sequences identified with at least two PSMs. Besides these filters, proteins were considered for further analysis when identified in at least 60% of the samples in at least one condition and only if they were regulated proteins. Statistical analysis was performed in Perseus, using Student’s *t*-test (*p* < 0.05). The mass spectrometry proteomics data have been deposited into the ProteomeXchange Consortium via the PRIDE [30] partner repository with the dataset identifiers PXD030200 (Username: reviewer_pxd030200@ebi.ac.uk Password: MthPoFtz) and PXD030199 (Username: reviewer_pxd030199@ebi.ac.uk Password: Ln8aUR61).

### 2.10. CBA Anaphylatoxin Assay

A multiplex cytometric bead assay kit (BD Biosciences) was used to measure the complement system anaphylatoxins (C3a, C4a, and C5a) according to manufacturer’s protocol. The analysis was performed in a two-laser BD FACSCalibur flow cytometer with CellQuest version 5.2 software (BD Biosciences). C3a, C4a, and C5a amounts (pg/mL) were calculated based on the standard curve and used to perform the statistical analysis (Mann–Whitney U-test).

### 2.11. Proteomics and Clinical Parameters Correlation

Statistically significant proteins from mice and humans were analyzed for degrees of correlation with several clinical parameters of malaria disease based on normalized abundance using Proteome Discoverer v.1.4 (Thermo Fisher Scientific, Waltham, MA, USA). The expression abundances were transformed into Log2 values, and the missing values were inputed from a normal distribution using a down-shift of 1.8 and a distribution width of 0.3, implemented in Perseus software v.1.6.10.43 [31]. Metaboanalyst tool (https://www.metaboanalyst.ca, accessed on 25 November 2022) was used to evaluate the correlation between regulated protein normalized abundances and clinical parameters using Spearman’s correlation coefficient at a *p*-value of < 0.05.

## 3. Results

### 3.1. Establishment of a Murine Translational Model of Malaria in Pregnancy for Biomarker Discovery

In this study, we developed a translational model to identify plasma biomarkers of MiP using samples collected from an in vivo murine model and human patients (Figure 1a,b). First, biochemical and molecular parameters were initially evaluated, followed by quantitative mass spectrometry-based proteomics of plasma samples and correlation analysis.

In the murine model, placental vascular space (%) was reduced (*p* < 0.0001) and spleen weight increased (*p* = 0.0286) in the infected group compared to the control group (Figure 2a, Appendix A). The reduced placental vascular space is hypothesized to be associated with lower blood flow in the placenta, resulting in decreased fetal viability and intrauterine growth restriction [29]. In the human cohort, clinical parameters of babies, such as newborn weight (*p* = 0.0106), head circumference (*p* < 0.0001), and chest circumference (*p* = 0.0063), displayed a significant decrease in the infected group compared to the human control group (Figure 2b, Appendix A).

During the erythrocytic stage, the spleen is involved in eliminating parasitized red blood cells and exhibiting the immune response [32]. Based on the physiopathology of MiP, it is expected to have higher spleen weight associated with malaria and lower placental vascular space related to birth defects. Moreover, reduced vascular space is a common placental alteration related to the infection during pregnancy [27]. In the murine model, placental malaria-associated changes include reduced vascular space due to the infection [33]. In the human placenta, placental malaria is characterized by histological and immune changes, such as monocyte infiltration and parasite pigment (hemozoin) deposition, which result in low birth weight [34,35]. Another outcome observed in the newborns of mothers that developed malaria during pregnancy was the reduction in the head circumference. The reduced newborn head circumference was classified as small head (head circumference below one standard deviation of the mean) [36] and microcephaly (head circumference below two standard deviations of the mean) [37]. The head size classification was performed based on the WHO child growth standards [38]. Infants’ head circumference below 33.2 and 32.7 was considered a small head for boys and girls, respectively, and below 31.9 and 31.5 as microcephaly for boys and girls, respectively [36,37].

No significant differences were observed in mouse proinflammatory cytokines. On the other hand, IL-10 showed a significant (*p* = 0.0030) increase in the infected group compared to the control group (Figure 3a–d, Appendix A). In human samples, IL-10 levels were also increased (*p* = 0.0424) in the infected group compared to the control group (Figure 3e–h, Appendix A).

Regarding the alterations in the cytokine levels during malaria infection, TNF-α was described as a cytokine associated with splenomegaly in mice during the blood stage of malaria infection [39]. Increased peripheral IL-10 levels are related to placental malaria and are described as an inflammatory biomarker in placental malaria [40]. IL-10 levels play an essential role in reducing the inflammatory response during malaria infection, and increased levels of IL-10 were identified in the plasma of patients with malaria-associated splenomegaly [41]. Many inflammatory cytokines and chemokines have been associated with MiP [7]. IL-10 is a cytokine involved in immunopathology and protection during malaria, reducing the inflammatory response. It has been identified to be elevated during malaria episodes both in the serum of non-pregnant and pregnant patients [42]. High levels of IL-10 are associated with a reduced ability to eliminate parasitemia [43]. However, there is limited understanding of the molecular mechanisms by which malaria induces fetal dysfunctions [27,43,44].

A bottom-up proteomics was performed in the peripheral plasma from murine and human datasets with malaria during pregnancy. A total of 279 and 249 proteins were identified in murine and human samples, respectively (Figure 4, Appendix A), of which 75 regulated proteins (28% of the total) and 76 regulated proteins (30% of the total) were present in murine- and human-infected groups compared to the control (Figure 4, Appendix A). Based on the gene name of the total proteins, an overlap of 103 proteins was identified in both organisms.

Regulated proteins were associated with proteins involved in regulatory and signaling pathways that are both common and unique between infected mice and humans. We focused on the complement system based on identifying the regulated proteins from the complement components in the infected groups compared to control groups in both organisms.

### 3.2. Complement System Activation in Malaria in the Pregnancy Context

Figure 5 gives an overview of complement system pathways and their components identified in murine and human datasets. All murine-regulated complement proteins are highlighted in yellow when they were identified exclusively in the mice dataset, and proteins highlighted in green were regulated exclusively in humans and the ones in orange were common between both species.

In the mouse dataset, regulated proteins from the alternative pathway such as Lipopolysaccharide Binding Protein (LBP), Complement Factor B, and Complement Factor H were identified as up-regulated in the infected group (Figure 5, Appendix A). The increased levels of Complement Factor H were correlated with severity biomarkers and susceptibility in severe malaria cases [45].

In the human dataset, the following components from the alternative pathway were regulated: Complement Factor H-related protein 1, Complement Factor H-related protein 2 (down-regulated in the infected group), and Complement Factor D (up-regulated in the infected group) (Figure 5, Appendix A). The Lectin pathway showed only Ficolin-2 up-regulated in the human-infected group and Mannose-binding protein C, which was up-regulated in the murine-infected group and down-regulated in the human-infected group.

Regarding the classical pathway, Complement C1q subunit A, Complement C1q subunit B, Complement C1q C, Complement C1s, and Complement C2 were up-regulated in the murine-infected group (Figure 5, Appendix A). Moreover, in the human dataset, Complement C1r subcomponent-like protein (C1RL) and Complement C4-binding protein (C4BP) were down-regulated in the infected group, while the Complement C4a was up-regulated in the infected group (Figure 5, Appendix A). Together, the down-regulation of the C4BP combined with the up-regulation of the C4a anaphylatoxin suggests an activation of the complement system.

To confirm the complement system activation, anaphylatoxin C3a, C4a, and C5a measurement in the human maternal peripheral plasma was performed using a CBA assay.

### 3.3. Anaphylatoxin Measurement Confirms the Activation of the Complement System

*Plasmodium*-infected peripheral plasma anaphylatoxins were validated using the orthogonal method to confirm the activation of the complement system. A CBA anaphylatoxin assay was used to measure C3a, C4a, and C5a levels (pg/mL) in the infected pregnant samples (n = 15) compared to non-infected samples (n = 15). Figure 6a,b showed a significant increase in the C3a and C4a anaphylatoxins (*p* < 0.0001 and *p* = 0.0005, respectively) in the infected group compared to the control group. Figure 6c represents C5a; however, no significant difference was observed for this anaphylatoxin between the infected and control groups (*p* = 0.3398).

Taken together, these data confirmed the complement system activation observed in the proteomics data (by decreasing complement system inhibitors and increasing C4a) during malaria in pregnancy at the time of delivery.

### 3.4. Proteomics and Clinical Data Correlation in MiP

Regulated proteins from the murine dataset were correlated with clinical parameters (cytokines, spleen weight, fetus weight, and placental vascular space). Statistically significant correlations (*p* < 0.05) were considered in the analysis, presented in the correlation matrix (Figure 7).

Significant correlations between clinical parameters and complement components were highlighted in Table 1. Complement C1q subunits (A, B, and C), Properdin, and Complement C8 chains (alpha and gamma) were positively correlated to spleen weight. Moreover, Complement C1q subunits (A and B) and Complement C8 chains showed a negative correlation to placental vascular space and fetus weight. Also, Properdin presented a negative correlation with placental vascular space. Properdin, C8 chains (alpha and gamma), and Complement C1qB showed a positive correlation to IL-10. The complete list of the data correlation is shown in Appendix A.

Spearman’s correlation was also performed to the human dataset to observe the association between the regulated proteins and clinical parameters, mainly the newborn data and/or outcomes resulting from malaria in pregnancy. A correlation Heatmap is represented in Figure 8.

In the human dataset, complement components were correlated with newborn outcomes, such as the positive correlation between C4b-binding protein beta chain, Complement C1r subcomponent-like protein, and Complement Factor H-related protein 2 (down-regulated proteins in the infected group) with newborn head circumference. On the other hand, Ficolin-2 and Complement C4a (up-regulated in the infected group) were negatively correlated with the newborn head circumference (Table 2). Moreover, placental malaria was positively correlated to microcephaly and small head circumference, and negatively correlated to newborn chest circumference and weight. The complete list of correlations is provided in Appendix A.

## 4. Discussion

This study analyzed the plasma proteomic signature of mice and humans with malaria in pregnancy. A large proportion of proteins were shared and regulated in both organisms. However, differences and similarities between the organisms were reported. Previous studies have shown that some clinical outcomes observed in humans are also observed in the murine model of MiP, such as maternal anemia, which was correlated with increased parasitemia during pregnancy [46], fetal loss [47], and reduced birth weight [27,28]. In the human cohort analyzed in this study, placental malaria was classified into four categories according to Bulmer classification based on the presence of the malaria pigment and parasites: non-infected (0), active infection (1), active-chronic infection (2), and past-chronic infection (3) [48]. However, in the murine model, female mice were cured with a sub-curative dose of chloroquine (chronic infection). Due to that, we assessed common and exclusive plasma markers regulated during MiP.

In prospective and retrospective cohort studies performed by Dombroski et al. [44] showed a prevalence of newborns (30.7% and 36.6%, respectively) with small head circumferences from mothers exposed to malaria in pregnancy and a prevalence of microcephaly in newborns from mothers infected by *Plasmodium falciparum* [44]. In our results, we observed a reduction in the newborn head circumference from mothers exposed to malaria during pregnancy. The fetus’s exposure to maternal infection results in disruptions to normal neurodevelopment, impairment of neurotransmitter functions, and cerebral atrophy. Ozawa et al. [49] administered pregnant mice with polyriboinosinic–polyribocytidilic acid (poly I:C), which decreased the body weight gain (*p* < 0.001) and the number of fetuses in the litter (*p* < 0.001). Moreover, poly I:C administration intraperitoneally during pregnancy causes maturation-dependent memory impairment in adult mice [49]. Among the main findings of our study were the elevated levels of peripheral plasma IL-10 in the infected group from both organisms. IL-10 is a cytokine involved in immunopathology and protection during malaria infection, reducing the inflammatory response, and has been increased during malarial episodes both in the serum of non-pregnant and pregnant patients [42]. Moreover, high levels of IL-10 were associated with a reduced ability to eliminate parasitemia [43]. IL-10 displays a protective role in the tissue from infection-mediated inflammation during infection, preventing severe malaria symptoms, such as anemia and organ damage [50]. 

However, there is limited understanding of the molecular mechanisms by which malaria infection induces fetal dysfunctions [23,26,29].

The complement system also displays a relevant role during malaria infection and pregnancy. In malaria, the complement system has been highlighted as a complex of soluble, heat-sensitive, and cell surface-associated proteins that participate in pathogen opsonization, pathogen lysis, and recruitment of phagocytes via downstream C3 complement deposition [51]. Its activation in healthy pregnancy was confirmed after anaphylatoxins measurement (C3a, C4a, and C5a), which was increased in the placental plasma of healthy pregnant women compared to non-pregnant women. The complement activation during pregnancy is an innate response for both mother and fetus protection, compensating for the decrease in adaptive immunity, which is common during pregnancy [51]. Previous studies have suggested that the complement system is altered during MiP in peripheral plasma, and there is also evidence for complement components locally in the placenta. The human trophoblast secretes C3 and C4 and expresses mRNA for C6-9 [51]. In normal pregnancies, C1q, C3d, C4, C6, C9, C4BP, and Factor H were identified in healthy placentas [10], showing a presence of host defense against pathogens. Circulating levels of complement components such as C3, C3a, C4a, C4d, C5a, sC5b-9, C9, and Factor H are increased in pregnant women than in non-pregnant women [51,52].

The complement system activation during MiP influences birth outcome, inducing increased inflammation and dysregulation of the angiogenic process [15,29,53]. In particular, C5a mediates placental vascular insufficiency, resulting in low birth weight [15]. Conroy et al. [21] observed a link between increased levels of C5a, altered levels of angiogenic factors, and low birth weight in women with *P. falciparum* infection during pregnancy. In C5a receptor-deficient pregnant mice, placental vascularity is reduced during malaria in pregnancy and is correlated with improved birth weight and fetal viability [15]. In MiP, the complement system is activated to neutralize parasite invasion, and the increased/excessive activation of the complement can result in impaired fetal development and survival. Complement components (C5a-C5R signaling) were related to cognitive deficits due to malaria infection exposure during pregnancy [54]. C5a has been associated with dysregulated synaptogenesis, neuroinflammation, and angiogenesis. Indeed, the complement activation can result in increased cytokine levels in the fetus’ brain, disturbing normal neurodevelopment [54]. Moreover, higher terminal complement complex (TCC) levels were associated with lower birth weight [55]. Several studies also associated the complement system with brain injury, showing evidence of complement classical pathway role in the central nervous system synapse elimination [56], neurodegeneration, and neurodevelopment [57]. In the present study, we identified other complement proteins regulated by placental malaria and associated with newborn outcomes resulting from malaria infection during pregnancy.

## 5. Conclusions

This study shows the application of quantitative proteomics strategies to identify molecular alterations in the plasma of pregnant women with MiP. Our study identified similarities and differences in the translational model of malaria in pregnancy. In both organisms, IL-10 levels were increased in the infected groups. Regarding the clinical parameters, both in the murine and human-infected groups, we could identify negative outcomes such as reduction in the murine placental vascular space and a reduction in the head and chest perimeter of the newborns exposed to malaria in pregnancy. The proteomics analysis showed the regulation of plasma proteins during MiP in both organisms, with some differences in the abundance of the complement inhibitory proteins. The differences identified in the proteomics data are related to the limitations of the model. The complement system cascade was activated during MiP, which was confirmed using the CBA anaphylatoxin assay. Correlation between regulated proteins with immune markers expression and pathological parameters associated with MiP allowed us to prioritize specific plasma protein markers.

## 6. Limitations of the Study

The diagnostic performances of plasma markers identified in this study need to be addressed in a more extensive and independent cohort. Moreover, the expression of these markers during gestation should be evaluated to infer their potential as early markers. The correlation between clinical and molecular markers shown in this study will be the first report on MiP to identify diagnostic and prognostic markers.

## Figures and Tables

**Figure 1 genes-14-01624-f001:**
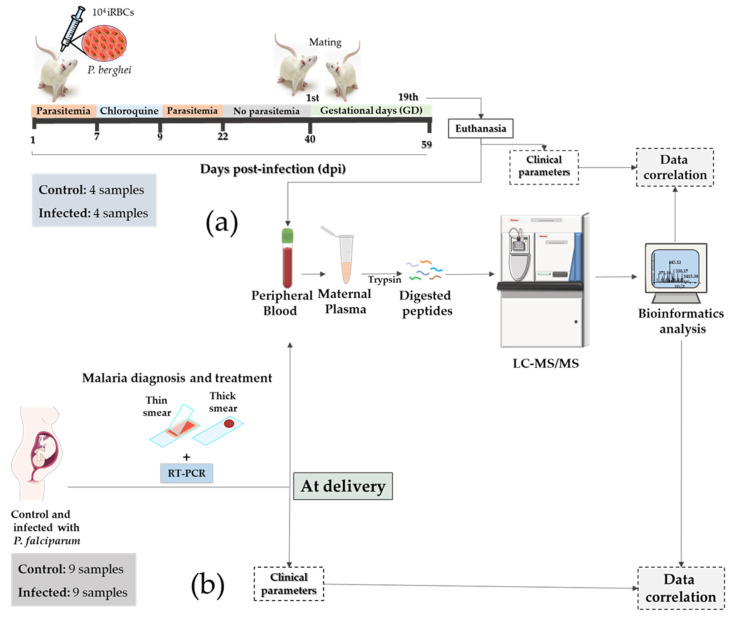
Comparative label-free proteomics approach in humans and mice with MiP. Translational-comparative approach for label-free proteomics in humans and mice with MiP. (**a**) Experimental workflow of the murine MiP model. BALB/c females were infected and treated with chloroquine (0.7 mg/animal) and were mated with BALB/c males at forty days post-infection. Pregnancy was detected by weight gain (3–4 g) at thirteen days post-mating. A cesarean was performed on the 19th gestational day, and peripheral blood was collected. Blood was collected in citrate tubes, and plasma was isolated using centrifugation to perform protein digestion (trypsin) and MS analysis. (**b**) Experimental workflow applied to the cohort of pregnant women that developed MiP. Pregnant women were diagnosed with malaria (gold standard—thin smear and PCR test). After malaria diagnosis, pregnant women were treated with antimalarial drugs, according to Brazilian Ministry of Health (MoH) guidelines. Then, blood from non-infected and infected women was collected, and plasma was isolated to perform protein digestion (trypsin) and then mass spectrometry analysis. No parasitemia was detected in the peripheral plasma at delivery. However, parasite molecules were detected in the placenta (hemozoin).

**Figure 2 genes-14-01624-f002:**
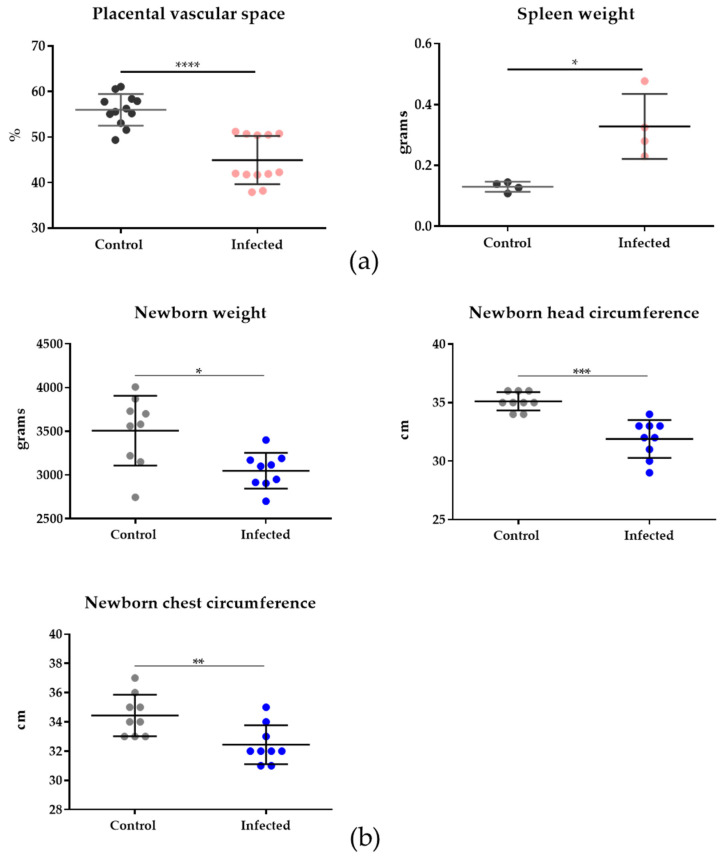
Clinical parameters of mice and humans with MiP. (**a**) Clinical parameters of mouse samples—placental vascular space (%) and spleen weight (g) were measured in control (n = 4) and infected (n = 4) groups. Placental vascular space (%) was significantly reduced (*p* < 0.0001) in infected (n = 12, corresponding to four animals) samples compared to the control group (n = 12, corresponding to four animals). Spleen weight was increased (*p* = 0.0286) in the infected group (n = 4) compared to the control group (n = 4), an expected effect of *Plasmodium berghei ANKA* infection. (**b**) Clinical parameters of the human dataset. Newborn weight (*p* = 0.0106), head circumference (*p* < 0.0001), and chest circumference (*p* = 0.0063) displayed significant differences between the infected (n = 9) and control (n = 9) groups. Mann–Whitney test (*p* < 0.05) was performed to compare infected and control groups. Statistical analysis was performed using GraphPad Prism version 6. The astherisks indicate statistical significance between the groups. For murine data, **** represents *p* < 0.0001, * is *p* = 0.0286. For human data, * represents *p* = 0.0106, *** for *p* < 0.0001, and ** *p* = 0.0063.

**Figure 3 genes-14-01624-f003:**
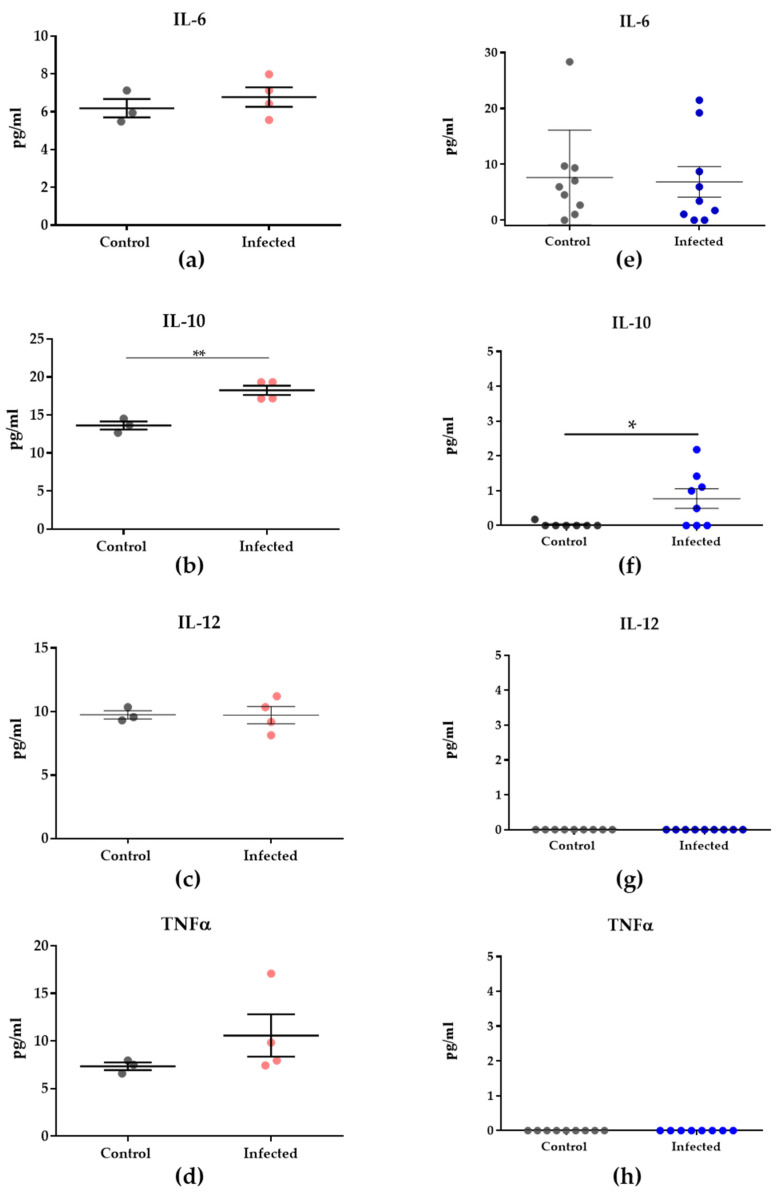
Cytokine levels of murine and human samples. (**a**–**d**) Cytokine levels of murine samples (n = 4 per group; however, control samples presented N/D values for some cytokines). (**e**–**h**) Human peripheral plasma (control = 9, infected = 9). All data were submitted to Robust Regression and Outlier Test (ROUT, Q = 0.1%), and Mann–Whitney was performed to evaluate differences between the two groups. Asterisks indicate statistical significance. Legend: Light pink circles represent the murine in-fected samples and dark blue circles correspond to the human infected samples. ***
*p* = 0.0424, ****
*p* = 0.0030.

**Figure 4 genes-14-01624-f004:**
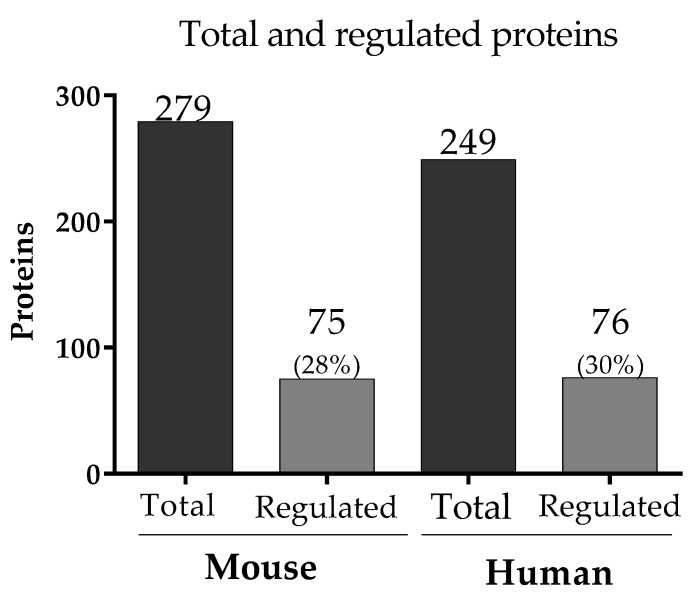
The number of proteins in murine and human datasets. In mouse and human samples, 279 and 249 proteins were quantified, respectively (represented in dark grey bars), and the regulated proteins in mice (28% of the total proteins) and humans (30% of the total proteins) are depicted in light gray.

**Figure 5 genes-14-01624-f005:**
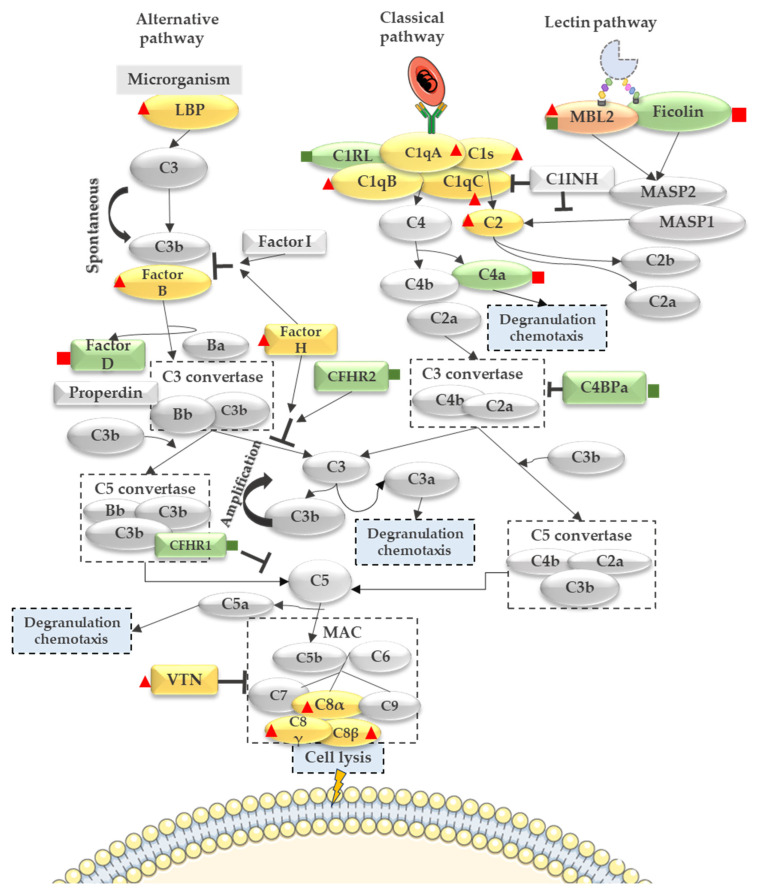
Complement system pathways and regulated proteins in the infected group. In the mouse dataset (yellow molecules), classical pathway complement components such as C1q subunits (A, B, and C), C1s, and C2 were up-regulated in the infected group (red triangle). Moreover, alternative pathway complement components were also up-regulated in the infected group (LBP, Factor B, and Factor H). Furthermore, in the MAC complex, Vitronectin (VTN) and C8 subunits (alpha, beta, and gamma) were up-regulated in the murine-infected group. In the human dataset (green molecules), Complement C1r subcomponent-like protein and C4-binding protein from the classical pathway were down-regulated (green square), and Complement C4a was up-regulated in the infected group (red square). From the alternative pathway, Complement Factor D was up-regulated (red square), while the complement inhibitors Complement Factor H-related proteins 1 and 2 were down-regulated in the infected group (green square). Regarding the Lectin pathway, Ficolin-2 was up-regulated in the infected group of the human dataset, and Mannose-binding protein C was identified in both organisms, being up-regulated in the murine-infected group (red triangle) and down-regulated in the human-infected group (green square).

**Figure 6 genes-14-01624-f006:**
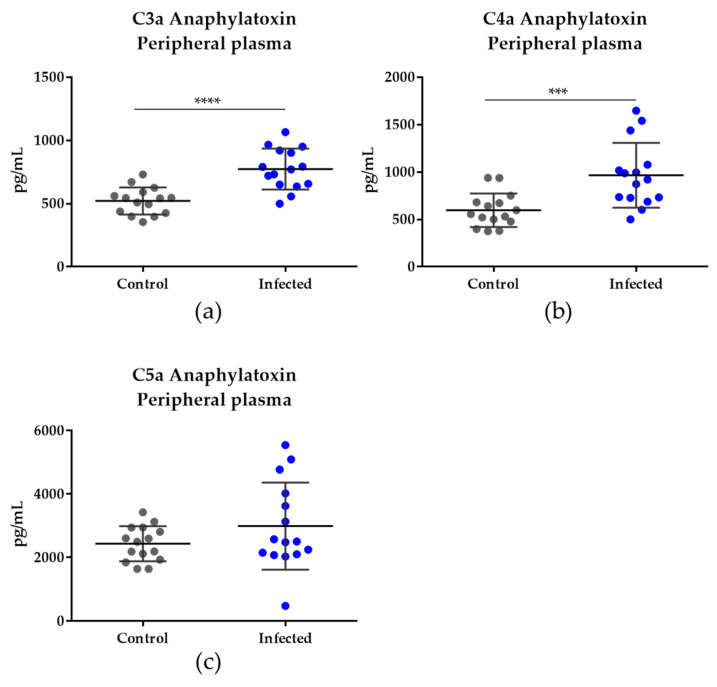
Anaphylatoxin measurement in the peripheral plasma of the infected pregnant women. (**a**) C3a anaphylatoxin showed a significant increase in the infected group compared to the control group (*p* < 0.0001). (**b**) C4a anaphylatoxin also increased in the infected group compared to the control group (*p* = 0.0005). (**c**) No statistically significant difference in C5a anaphylatoxin level was detected between the infected and control groups. Astherisks indicate the statistical significance. **** represents *p* < 0.0001 and *** *p* = 0.0005.

**Figure 7 genes-14-01624-f007:**
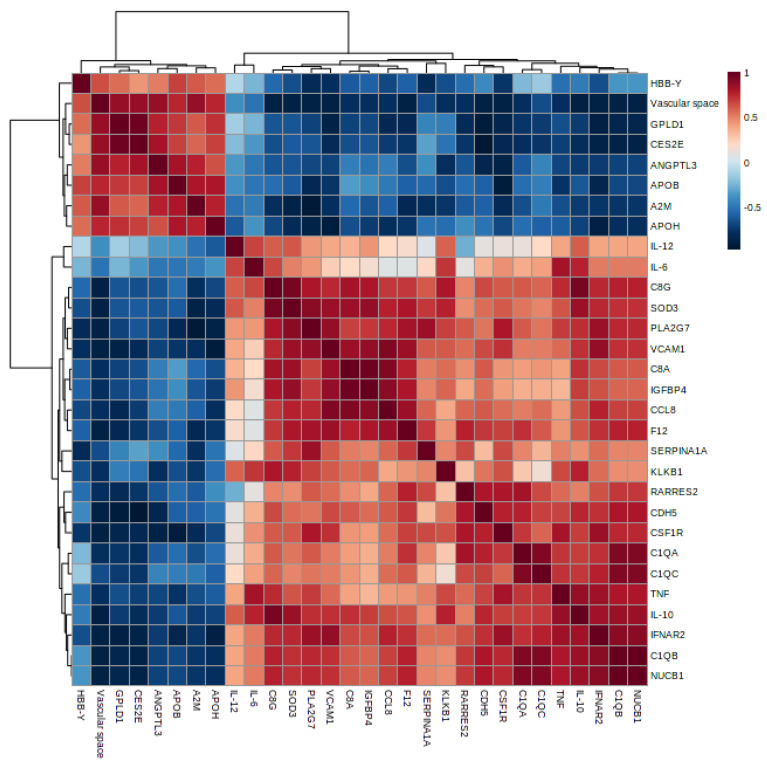
Matrix of the significant Spearman’s correlation from murine-regulated proteins and clinical parameters.

**Figure 8 genes-14-01624-f008:**
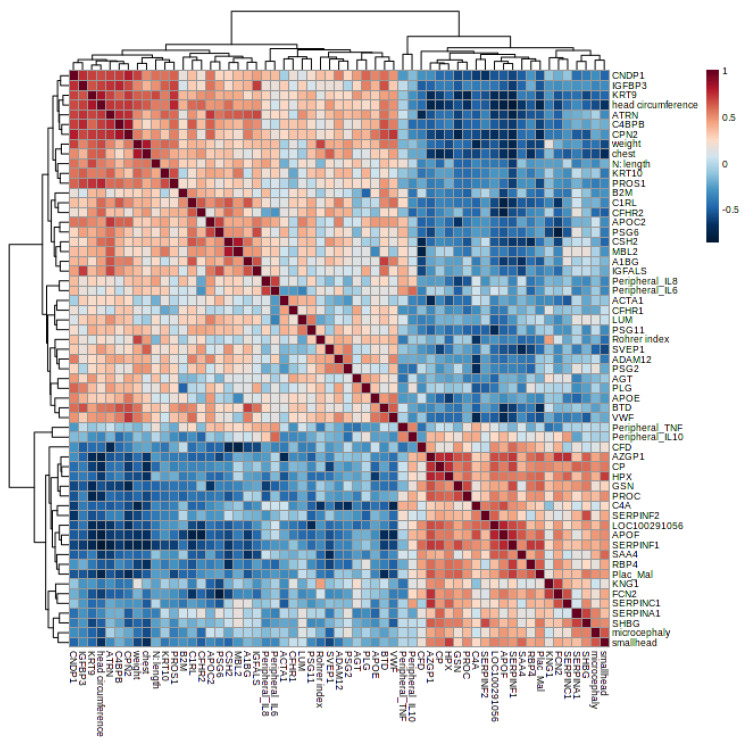
Matrix of the significant Spearman’s correlation of human regulated proteins and clinical parameters.

**Table 1 genes-14-01624-t001:** Significant Spearman’s correlation between murine-regulated proteins from complement component and clinical parameters.

Protein ID	Protein Name	Placental Vascular Space	Fetus Weight	Spleen Weight	IL-10
P98086	Complement C1q subcomponent subunit A	−0.76376	−0.76376	0.78571	
P14106	Complement C1q subcomponent subunit B	−0.87287	−0.87287	0.92857	0.74253
Q02105	Complement C1q subcomponent subunit C			0.78571	
Q8K182	Complement component C8 alpha chain	−0.76376	−0.76376	0.78571	0.74253
Q8VCG4	Complement component C8 gamma chain	−0.87287	−0.87287	0.7619	0.81439
P11680	Properdin	−0.87287		0.92857	0.77846

**Table 2 genes-14-01624-t002:** Significant Spearman’s correlation between human-regulated proteins from complement component and clinical parameters.

Protein ID	Protein Name	Placental Malaria	Newborn Weight	Newborn Head Circumference	Newborn Chest
P20851	C4b-binding protein beta chain	−0.5906		0.72381	
Q9NZP8	Complement C1r subcomponent-like protein	−0.47703	0.50891	0.60161	
Q15485	Ficolin-2			−0.59848	
P36980	Complement Factor H-related protein 2			0.6225	0.48617
P0C0L4	Complement C4-A		−0.64629	−0.47836	−0.58152

## Data Availability

Raw mass spectrometry proteomics data have been deposited to the ProteomeXchange Consortium via the PRIDE (https://www.ebi.ac.uk/pride, accessed on 6 December, 2021) partner repository with the dataset identifiers PXD030200 (Username: reviewer_pxd030200@ebi.ac.uk Password: MthPoFtz) and PXD030199 (Username: reviewer_pxd030199@ebi.ac.uk Password: Ln8aUR61).

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
