# Peer review of "Complement System Activation Is a Plasma Biomarker Signature during Malaria in Pregnancy"

_genes, 2023, doi:10.3390/genes14081624_

Round 1

Reviewer 1 Report (New Reviewer)

The study by Santiago et al. reports on the identification of serum markers associated with complement activation during malaria infection in pregnancy. The subject is relevant and timely.

1.      Provide an explanation for the initiators of the classical complement pathway during pregnancy in malaria infected women. With reduced adaptive immune responses, are parasite specific antibodies elevated during pregnancy to activate the classical complement pathway?

2.      Section 2.7 indicates a sample size of 10 µl of plasma proteins (containing 100 µg protein?) for proteomic analysis. Is this reproducible? How many times was this analysis performed? What was the coverage each time the analysis was performed? What were the controls?

3.      Further explanation is required for the correspondence of the mouse and human data regarding complement components and cytokines, in particular the significance of elevated IL-10 levels and the clinical parameters in both studies.

4.      In figures 6 and 7, parameters measured are different in mouse and human samples, and in the clinical parameters characterized (see #3) such as head circumference of newborns, reduced birth weight etc. An explanation should be added in the discussion to specify the differences, e. g. effects of different treatment times in pregnant women compared to the treatment times in mice.

5.      In the conclusion, “similarities and differences” observed in mice and human studies should be mentioned and the specific protein markers also mentioned.

6.      Manuscript should be checked for grammar.

7.      Check for consistency of font size throughout the manuscript.

Minor editing of grammar required.

Author Response

Please see attachments.

Reviewer 2 Report (New Reviewer)

A few major issues, 

1) why were the mice infected, treated, then mated, but not infected during pregnancy if this is about MiP?

2) why were parasitaemia not taken during pregnancy? How would we know if the parasites were present?

3) Mice were infected prior to pregnancy, it is not surprise that spleen is larger in Fig2

4) what were the inflammation markers before mating in infected/treated animals?

I am not convinced that the mouse experiments were representative of MiP

minor edits

Author Response

Please see attachments.

Reviewer 3 Report (New Reviewer)

Santiago et al. evaluate the proteomic studies to study the difference of serum protein between control and malaria in pregnancy cohort in either mice and human. They found the complement system activation is associated with Mip and poor outcome of fetus. 

The approach to find out the potential biomarker is novel and important for either prevention or treatment for MiP. But the complement activation has been characterized before as the introduction and this study did not bring more information beyond that. There are couple of the issues about the statistics and presentation as noted below. 

For vascular space, the original paper use "The blood vascular area in each placenta was estimated from the analysis of two non-consecutive sections. The reported results correspond to individual pregnant females and represent the average result for 2-3 placentas.” In this study, the authors used: n=12, corresponding to 4 mice to calculate the statistics in Fig 2A. This is not appropriate for comparison.  

For statistics method, the authors used “t test following Mann-Whitney posttest” (Fig 2, 3) and there was no statistics method in Figure 6. As searching for “t test following Mann-Whitney posttest”, there is no information about this. The authors need to clarify the methods and the reason for choosing the method otherwise the results are difficult to interpretate. 

The 1st paragraph of page 11 may be more appropriate in the discussion. 

In figure 6 (Page 14), based on the supplement result, we know that the authors used sperate participants for validation the difference. It should be emphasized in the text that separate participants were chosen for this. And there should be some reasons that why these 15 infected or uninfected were chosen. The same reason we should know about how to pick up those 9 control and 9 infected proteomic study the group. 

In Fig7 and 8 , more detail explanation and label about figures are needed for the reader to understand. It is better to use gene names instead access peptide ID. 

Since the author measured the parasitemia in the mice, the parasitemia status before and chloroquine treatment will be helpful for the reproduce of the studies. Besides, the authors mentioned “ chronic infection”, the status of parasitemia will be need to support this statement.  

For CBA anaphylatoxin assay, it should be able to show the exact concentration of each C3a, C4a and C5a instead of MFI. 

Round 2

Reviewer 1 Report (New Reviewer)

The response to comments #1 regarding initiators of the complement pathway during pregnancy should be incorporated into the introduction section  with the citations added to the list of references.

The response to comments #2 regarding sample size and protein concentration should be added to section 2.7.

The response to comment #3 regarding IL-10 should be added to the discussion section and the citations added to the list of references.

Author Response

Thanks to reviewer 1 for the comments. The response to comments #1 regarding initiators of the complement pathway during pregnancy should be incorporated into the introduction section  with the citations added to the list of references. OK The response to comments #2 regarding sample size and protein concentration should be added to section 2.7. The protein concentration (100ug) is already in section 2.7. The sample size is in sections 2.3 and 2.5 for mice and humans, respectively.   The response to comment #3 regarding IL-10 should be added to the discussion section and the citations added to the list of references OK

Reviewer 2 Report (New Reviewer)

None

Author Response

Thanks to reviewer 2.

Reviewer 3 Report (New Reviewer)

The authors have answered all my concerns. 

Author Response

Thanks to reviewer 3.

This manuscript is a resubmission of an earlier submission. The following is a list of the peer review reports and author responses from that submission.

Round 1

Reviewer 1 Report

I thank the editorial team for providing me with the opportunity to review the manuscript submitted to the journal Genes under the title "Complement system activation is a plasma biomarker signature during malaria in pregnancy". I credit the authors for having identified the area that is of high importance in the field of malaria and having been able to chart a compelling study for the readers. The authors have managed to compile an impressive set of samples of MiP samples and in addition, develop a murine model for the condition making the study a relatively novel and impressive one.

Having said that, it is with extreme sadness I would like to point the major concerns associated with the study that needs to be addressed before  publication. My utmost concern is regarding the proteomic analysis of the samples which forms the basis of the conclusions made.

1.     It is difficult understand the decision made by the authors to compare the peptide abundances as opposed to the protein abundances. There are several concerns in this aspect

a.     There is a poor coverage of proteins in the study from each sample. On an average we identify at least about >1000 proteins in a plasma sample from a orbitrap analysis. Several studies have outlined the plasma proteomes in various conditions and compared to them, the majority of the proteins identified in the study seem to be limited largely to the abundant proteins in the current study. There are several guidelines and quality check tools to analyze the authenticity of biomarker discovery from the plasma proteomes such as "Plasmaproteomeprofiling.org" by a leading proteomics group. It could be useful to compare the data from this study against the control datasets to eliminate some of the apparent contaminants and assess the quality of the dataset before making the calls of biomarker discovery.

b.     It is not clear as to why the authors chose to perform multiple label free runs on multiple platforms as well. Perhaps a labeled quantitative analysis using a 10 -plex TMT would have been better or at least a DIA evaluation could be considered. The results presented here are not ideal to make any evaluations considering the fact that these are only the most commonly found abundant proteins in plasma samples. Adding to the complexity is the fact that multiple runs bring in additional variables owing to the status and conditional variables of the mass spectrometer and is inherently sub-par to provide meaningful data for comparison. 

c.     I am unable to understand the rationale behind the use of peptide abundance comparison since the variation in abundances of the multiple peptides of any given protein only highlights the disparity in such a study. It is evident in the fact that the differentials noticed in the peptides are not true for their corresponding proteins. For example, the authors state that they find complement C1q subcomponent B to be upregulated in the infected group vs the control. Upon examining the supplementary data, I find that there are 7 peptides that are identified for this C1q subcomponent B, of which only 2 peptides are upregulated in infected samples while the rest 5 are basically downregulated. In such a case it is often the unique identity of the peptides for the protein grouping that is utilized for determining the differential. However, in the example all of these peptides are assigned to a single protein. This makes it hard to consider the actual protein level in the infected sample compared to the control and hence to conclude any meaningful inference and use of it as a biomarker. Proteins are the components/players of the differentially regulated highlighted in the study and not the individual peptides. So, it is hard to see the corelation between the results presented with the inferences made.

2. In the murine model, seems like the parasitemia was cleared before mating (Fig1). Therefore, it is not clear as to how this could serve as the model for gestational infection and comparable to human samples

3. There are several discrepancies in the number of human samples and controls considered. In the methods section 2.5, the authors state that “For the proteomic analysis, a total of ten pregnant women, five control and five infected by P. falciparum, were selected.” However, in the result section, the cytokine measurements show a n of 25 for controls and 334 for infected while the proteomic data in the supplementary files show data of 9 controls and 10 infected. It is extremely confusing to identify the sample cohorts used and their co-relations. At best, these samples seem too random and is hard to draw any correlation between the physiological parameters, cytokine levels and proteomic analyses.

Similar confusion extends to the data depicted in the newborn cohort. I urge the authors to clearly state the cohorts used and make it easy on the reader to draw proper correlation between the clinical parameters and data with the proteome profiles. As it stands this is extremely hard to understand.

4. In Figure 3A-D, it is not clear as to what the cytokine levels correspond to? Peripheral plasma of the mice or placental samples? If peripheral plasma, could the authors explain as to the reason for choosing to show only 3 control and 4 test samples while the corresponding proteome was profiled for 5 each? What happens to the cytokine levels of these 5 samples used for proteomics?

With reference to the clinical parameters of the human cohort provided in the supplementary table 2, they do not seem to be statistically significant. In addition, there were a few clear outliers as is the case with Sample 5 of the infected group. This raises questions as to why these were selected of the many samples available for proteomic analyses.

Reviewer 2 Report

The manuscript of Santiago et al entitled ‘Complement system activation is a plasma biomarker signature during malaria in pregnancy’ reports the findings of a label free quantitative comparative proteomics study of changes to the plasma proteome from malaria infection during pregnancy in humans and mice.

I think there are serious methodological flaws in the experiments that need to be addressed by the authors before the manuscript can be considered for publication.

Major points

1)           A power calculation for sample size is missing. A sample size of 5 samples for control and for patient samples seems low.

2)           Coverage of the plasma proteome is relatively low with 249 quantified human proteins compared to the human plasma proteome of 5877 proteins (Deutsch et al  Proteome Res. 2021, 20, 12, 5241–5263). The primary reason for the relatively low coverage is the experimental design that is missing a depletion strategy for highly abundant plasma protein. This likely explains the absence of detection of the cytokines displayed in figure 3.

3)           Mass spectrometry data has been acquired on two different LC-MS/MS platforms with different Orbitrap mass spectrometers (LTQ Velos and Q exactive) and different chromatographic conditions in particular different gradients and duration. Data has been collected on both systems with different quantitative outcomes for abundance ratio’s between infected and control. This is not addressed in the manuscript and no rationale is provided for analysing samples on two different platforms. For example, in the mouse dataset Complement factor I (Q61129) was quantified by 23 peptides in the Q exactive with an average log2 abundance ratio of 0.069 while it was quantified by 19 peptides in the LTQ Velos with an average log2 abundance ratio of -0.14. These are contradicting results and ignored in the manuscript.

4)           In my opinion there is too much focus on regulated peptides in the quantitative comparisons. These peptides are generated by bottom up proteomics during digestion with trypsin and should be mapped back to proteins before assessing quantitative differences in protein abundance levels between negative controls and samples. The example of Complement factor I (Q61129) shows that both upregulated and downregulated peptides were identified in a single protein by both mass spectrometers. It is not clear how this has been taken into account when discussing results at the protein level.

5)           The label free quantification method that is utilised in the manuscript is not described well, and I think there is some conflicting information about the quantitative analysis. It mentions peptide spectrum matches (PSMs) in section 2.9 and normalised abundance from Proteome Discoverer in section 2.10 which are different types of label free quantitative methods. The protein abundance data is processed using Perseus, which could indicate that LFQ data has been generated by a preceding MaxQuant analysis. Could the authors clarify which label free quantification method was used with reference to the study that introduced the method and how protein abundance values were generated.

 Minor points

 6. Introduction. ‘The plasma proteome of a P. berghei-infected pregnant mice was initially analyzed using quantitative mass spectrometry-based proteomics.’  Unclear what is meant with initially as this is a mass spectrometry-based proteomics study

7. Section 2.1 Number of samples (n)  is absent

8. Section 2.6 Volume of blood collected is missing

9. Section 2.7. Which protein assay was used to quantify the amount of plasma proteins

10. Figure 1 is missing the number of samples analysed and information about how proteins were identified and quantification

11. Figure 2. I think there are more dots in some of the figures than is mentioned in the figure legend (n=5 and n=12)

12. Figure 3 legend is missing description of method for detecting cytokine levels

13. Figure 4 should also present number of regulated proteins (or protein groups)